# Eosinophilic Granulomatosis with Polyangiitis: Latest Findings and Updated Treatment Recommendations

**DOI:** 10.3390/jcm12185996

**Published:** 2023-09-15

**Authors:** Ryu Watanabe, Motomu Hashimoto

**Affiliations:** Department of Clinical Immunology, Osaka Metropolitan University Graduate School of Medicine, Osaka 545-8585, Japan

**Keywords:** antineutrophil cytoplasmic antibody, eosinophil, eosinophilic granulomatosis with polyangiitis, interleukin-5, mepolizumab

## Abstract

Eosinophilic granulomatosis with polyangiitis (EGPA) causes necrotizing vasculitis and eosinophil-rich granulomatous inflammation in small- to medium-sized vessels, resulting in multiple organ damage. EGPA is classified as an antineutrophil cytoplasmic antibody (ANCA)-associated vasculitis, with myeloperoxidase-ANCA detected in approximately one-third of the patients. Conventional treatment of EGPA relies on systemic glucocorticoids (GCs) in combination with cyclophosphamide when poor prognostic factors are present; however, the dilemma between disease control and drug-related adverse effects has long been a challenge. Recent studies have revealed that the genetic background, pathophysiology, and clinical manifestations differ between ANCA-positive and ANCA-negative patients; however, mepolizumab, an interleukin (IL)-5 inhibitor, is effective in both groups, suggesting that the IL-5-eosinophil axis is deeply involved in the pathogenesis of both ANCA-positive and ANCA-negative EGPA. This review summarizes the latest knowledge on the pathophysiology of EGPA and focuses on the roles of eosinophils and ANCA. We then introduce the current treatment recommendations and accumulated evidence for mepolizumab on EGPA. Based on current unmet clinical needs, we discuss potential future therapeutic strategies for EGPA.

## 1. Introduction

Eosinophilic granulomatosis with polyangiitis (EGPA) is a necrotizing vasculitis that affects small- and medium-sized blood vessels [1]. EGPA is histopathologically characterized by extensive infiltration of eosinophils and causes organ damage due to tissue inflammation and ischemia [2]. EGPA was first reported by Churg and Strauss in 1951 [3] and is now recognized as an antineutrophil cytoplasmic antibody (ANCA)-associated vasculitis [4]. EGPA is also embedded within the scope of hypereosinophilic syndromes [5]. The annual incidence and prevalence of EGPA are 0.9–2.4 and 10.7–17.8 per million persons, respectively [6], and the disease occurs most frequently in people aged 40 to 60 years old [7].

The development of EGPA is divided into three stages. The first stage is the prodromal stage, characterized by bronchial asthma, allergic rhinitis, and sinusitis, and usually lasts for 3–10 years. The second stage is the eosinophilic stage, in which eosinophilia and eosinophilic infiltration into tissues and organs can be observed. The third stage is the vasculitic stage, which demonstrates clinical manifestations consistent with vasculitis, such as palpable purpura and peripheral neuropathy [4,7]. However, these stages may overlap in some patients.

Myeloperoxidase (MPO)-ANCA is detected in 30–35% of patients with EGPA [8]. Recently, it has been reported that the pathophysiology of EGPA can be divided into two disease phenotypes based on ANCA positivity: an ANCA-positive vasculitis-driven condition and an ANCA-negative eosinophil-driven condition [6,9]. ANCA positivity is associated with a higher frequency of glomerulonephritis and peripheral neuropathy, whereas myocarditis and lung involvement are often observed in ANCA-negative patients [10,11].

Glucocorticoids (GCs) are the mainstay treatment for EGPA [12,13,14]. Cyclophosphamide is recommended as an immunosuppressant in patients who do not respond well to GC therapy or have poor prognostic factors defined by the five-factor score (FFS) that predicts the severity and prognosis of EGPA [15,16]. In patients refractory or intolerant to cyclophosphamide, rituximab, mycophenolate mofetil, and azathioprine are alternatively used [17]. Mepolizumab, an interleukin (IL)-5 inhibitor, has been increasingly used for the treatment of EGPA based on a recent clinical trial [18]. However, because evidence regarding the use of mepolizumab for acute severe EGPA is still lacking, there remains an unmet need in the treatment of EGPA [19].

This review summarizes the current evidence on the pathophysiology of EGPA and highlights the roles of eosinophils and ANCA. We then introduce recent advances in the treatment of EGPA, focusing on the current treatment recommendations and therapeutic targeting of IL-5. Finally, we discuss the unmet clinical needs of EGPA. Through this process, we explore the potential therapeutic strategies for EGPA.

## 2. Physiological and Pathological Role of Eosinophils

### 2.1. Maturation and Migration of Eosinophils

The precursors of eosinophils are derived from CD34+ hematopoietic stem cells, which differentiate into mature eosinophils in bone marrow (Figure 1). They differentiate under the instructions of transcription factors, such as GATA-1 and PU.1, and mature in the presence of cytokines, such as granulocyte macrophage-colony-stimulating factor (GM-CSF), IL-5, and IL-3. Of these cytokines, IL-5 is the most specific to the eosinophil lineage [20]. After maturation, they migrate into the blood and circulate throughout the body. Mature eosinophils are not normally present in other organs except in the thymus, spleen, lymph nodes, uterus, and gastrointestinal tract from the stomach to the large intestine [21,22].

### 2.2. The Essential Role of IL-5 in Eosinophil Maturation and Survival

IL-5 is essential for eosinophil maturation and functions as a survival factor in mature eosinophils [23]. In fact, IL-5-transgenic mice, wherein IL-5 is overproduced by thymocytes and T cells, show massive eosinophilia [24]. These mice develop extramedullary eosinophilopoiesis and eosinophil infiltration in nearly every organ system. They also display severe inflammatory pathologies, such as ulcerating skin lesions and lower bowel inflammation [24], which resemble the clinical features of human EGPA. At steady state, IL-5 promotes eosinophil migration into restricted sites, such as the thymus, spleen, lymph nodes, and uterus; however, during inflammation, it mobilizes eosinophils from the bone marrow to various tissues and organs, which further enhances tissue inflammation [20,21].

### 2.3. Extravasation of Eosinophils

During allergic inflammation, eosinophils in the peripheral blood adhere to the vascular endothelium, roll around, and leak out of the vessels (Figure 1). Selectins and integrins, adhesion receptors expressed on eosinophil cell membranes, play important roles in this process [25]. Selectins help eosinophils bind to ligands on the vascular endothelium. Subsequently, β1 and β2 integrins expressed in eosinophils, respectively, bind to vascular cell adhesion molecule-1 (VCAM1) and intercellular adhesion molecule-1 (ICAM1) expressed in vascular endothelial cells, promoting eosinophil extravasation [20,25,26]. Extravasated eosinophils migrate to inflammation sites through the cooperation of IL-5 and eotaxins produced by epithelial and stromal cells. In addition, eosinophil trafficking to inflamed tissues involves multiple chemokines, such as CCL11 and eotaxin-3 (CCL26), and cytokines, such as IL-4, IL-5, and IL-13 [20,21].

### 2.4. The Emerging Role of ILC2s on Eosinophil Tissue Damage

As previously mentioned, IL-5 plays a pivotal role in eosinophil extravasation. IL-5 is mainly produced by T helper 2 (Th2) cells during allergen-induced acquired immunity [23]. Eosinophils differentiated by IL-5 produce IL-25 and further promote IL-5 production by Th2 cells [27]. Recent studies have demonstrated that type 2 innate lymphoid cells (ILC2s) can be an alternative source of IL-5 in innate immunity [23]. Specifically, ILC2s produce IL-5 in response to IL-33, which is released from endothelial and epithelial cells, such as the airway epithelium, in response to viral and bacterial infections.

Innate lymphoid cells (ILCs) do not have antigen receptors and thus cannot respond to antigens specifically but are involved in mucosal immunity and tissue repair by releasing large amounts of cytokines during the initial immune response [28,29,30]. Recent studies using animal models have revealed an essential role of the ILC2s in the pathogenesis of EGPA as well as allergic diseases [31,32].

A study conducted by Jarick et al. demonstrated that Neuromedin U receptor 1 (Nmur1) is specifically expressed in ILC2s [31]. The authors generated Nmur1-knockout mice, which showed a specific deficiency in the ILC2s; compared to control mice, these mice lacked IL-5 in their tissues, such as the lungs and intestines, resulting in reduced eosinophil infiltration even when allergic asthma was experimentally induced. These results strongly suggest that ILC2-derived IL-5 is essential for eosinophil tissue damage [31].

Another study conducted by Kotas et al. showed that using the previously mentioned IL-5-transgenic mice [24], administration of IL-33 caused an alveolar hemorrhage in these mice, and the alveolar fluid was enriched with eosinophils [32]. However, when IL-5-transgenic mice were crossed with ILC2-deficient mice, alveolar hemorrhage was not observed after IL-33 administration, indicating that IL-33-induced ILC2 activation is required for eosinophil alveolar damage. Additionally, the authors showed that circulating ILC2s decreased in patients with active EGPA compared to healthy controls, suggesting that ILC2s migrated into tissues during the active phase of EGPA [32].

Taken together, these studies revealed that ILC2-derived IL-5, rather than Th2-derived IL-5, plays an essential role in eosinophil tissue damage and that ILC2s could be a potential therapeutic target for EGPA.

### 2.5. Tissue Damage Caused by Eosinophils: Degranulation and Cytotoxicity

Circulating eosinophils in the peripheral blood are at a resting state; however, once they migrate into target tissues, they initiate degranulation in allergic diseases [23]. A characteristic feature that distinguishes eosinophils from other granulocytes, such as neutrophils and basophils, is the presence of large specific granules. Granules are cationic proteins, and their main substances include major basic protein (MBP), eosinophil cationic protein (ECP), eosinophil peroxidase (EPO), and eosinophil-derived neurotoxin (ENT) [25]. They are cytotoxic and are each associated with different organ damage. For example, MBP is involved in airway remodeling, asthma, fibrosis, and hypercoagulation, whereas ECP is associated with cardiotoxicity, hypercoagulation, fibrosis, and nerve fiber degeneration. ENT causes neurotoxicity, whereas EPO cooperates with NADPH oxidase to produce reactive oxygen species (ROS), thereby contributing to vascular endothelial damage and coagulation activity. All these mechanisms of eosinophil-driven endothelial cytotoxicity and/or pro-coagulant effects can lead to various clinical pictures of eosinophil-driven cardiovascular toxicity, including venous thromboembolism [33,34], thromboangitiis obliterans-like disease, [35] and even eosinophilic vasculitis [36].

There are three major mechanisms underlying eosinophil degranulation: exocytosis, piecemeal degranulation, and cytolysis. During exocytosis, the entire granule contents fuse with the cell membrane and are released, which is rarely observed in vivo. Piecemeal degranulation can selectively release various proteins via secretory vesicles. Cytolysis is the release of cytoplasmic proteins and intact eosinophil granules, and cytolytic degranulation occurs in 30–80% of eosinophils in inflamed tissues [23].

### 2.6. New Form of Degranulation: Eosinophil Extracellular Traps

Recently, the cytolytic degranulation of eosinophils has been shown to be an active cell death program rather than passive apoptosis. This form of cytolysis is called eosinophil extracellular trap cell death (EETosis) because, similar to neutrophil extracellular trap cell death, EETosis forms a structure that extrudes reticular DNA extracellular traps [37].

Eosinophil extracellular traps consist of mitochondrial DNA scaffolds and chaotic granule proteins. They are a part of the innate immune response and exhibit bactericidal properties [38]. Eosinophil extracellular traps are involved in the pathogenesis of eosinophilic diseases, such as asthma, allergic pulmonary aspergillosis, and eosinophilic sinusitis [23].

A more recent study showed that EETosis can be observed in eosinophils infiltrating the inflamed tissues of patients with EGPA [39]. Galectin-10, which accounts for 10% of eosinophil cytoplasmic proteins, is released extracellularly by EETosis. Galectin-10 forms Charcot–Leyden crystals, which are a typical feature of eosinophilic inflammation. Serum galectin-10 levels are positively correlated with serum IL-5 levels and the Birmingham vasculitis activity score (BVAS), which indicates the disease activity of EGPA [40]. Thus, galectin-10 may be a surrogate marker for EETosis and disease activity in EGPA [39]. However, the limitation of these studies is that phorbol 12-myristate 13-acetate-stimulated EETosis in vitro is not physiological, and it is desirable to confirm the extracellular release of reticular DNA using live cell imaging.

### 2.7. Differential Diagnoses of Blood Eosinophilia

Eosinophils in the peripheral blood normally range from 50 to 500/μL, but >500/μL is defined as blood eosinophilia and >1500/μL as hypereosinophilia [22]. In the newly proposed classification criteria for EGPA, a blood eosinophil count >1000/μL is specified [41]. Relatively common differential diagnoses of blood eosinophilia include parasitic and fungal infections, allergies including drug allergies, and atopic diseases. Hematological disorders (e.g., eosinophilic leukemia and malignant lymphoma), solid tumors (e.g., lung and colon cancer), chronic graft-versus-host disease, chronic inflammatory disease (e.g., inflammatory bowel disease), and autoimmune diseases are relatively rare but should be considered [42,43,44].

Hypereosinophilic syndrome (HES) is defined as a peripheral blood eosinophil count >1500/μL and organ damage is due to hypereosinophilia [22]. However, the organ damage caused by HES is not easily distinguished from that caused by EGPA, rendering HES a potential mimic of EGPA [45]. Biomarkers and scoring systems for differentiating HES from EGPA, such as eotaxin-3, E-CASE score, and C-reactive protein, have been reported [46,47,48,49].

## 3. Pathophysiological Differences between ANCA-Positive and ANCA-Negative EGPA

### 3.1. Clinical Manifestations Differ Based on the ANCA Status

MPO-ANCA is detected in 30–35% of patients with EGPA [8], and clinical manifestations differ based on ANCA status [2,19] (Figure 2). In ANCA-positive cases, vasculitis is the main pathology, with a high prevalence of rapidly progressive glomerulonephritis and peripheral neuropathy, whereas eosinophil infiltration is the predominant pathology in ANCA-negative cases, with myocarditis and pulmonary involvement being more frequent. This section outlines the differences between ANCA-positive and ANCA-negative pathophysiology. We then explain the target organs and histopathological differences between the two groups. Finally, we introduce a recent genetic study that revealed the mechanisms underlying pathological differences based on ANCA status [2,4,6,8,9,12,50,51,52,53].

### 3.2. Pathophysiology in ANCA-Positive and ANCA-Negative EGPA

In ANCA-positive EGPA, neutrophils and B cells play critical roles in the pathogenesis (Figure 3). Circulating neutrophils are primed by inflammatory cytokines, such as tumor necrosis factor and IL-1β, and C5a complement factor. Primed neutrophils then expose ANCA antigens normally encapsulated in cytoplasmic granules to the cell surface, allowing circulating ANCA to bind to ANCA antigens. The Fc fragments of ANCA activate neutrophils by interacting with FcγRIIa and FcγRIIIb. Activated neutrophils not only release ROS and cytotoxic enzymes but also form neutrophil extracellular traps (NETs), which result in vascular injury. Prolonged MPO exposure caused by NETs formation ultimately stimulates ANCA production by activated B cells, resulting in a vicious cycle [2,6,8,52].

In contrast, in ANCA-negative EGPA, Th2 cells, ILC2s, and eosinophils play a central role (Figure 3). As described previously, IL-5 released from Th2 cells during adaptive immunity and ILC2s in response to IL-33 during innate immunity promote eosinophil maturation, proliferation, and extravasation [12,20,21,23,25]. Extravasated eosinophils are recruited to inflammatory sites through the concerted action of IL-5 and eotaxins and degranulate when they reach target tissues [21,23]. Cytosolic granules, including MBP, ECP, EPO, and ENT, are cytotoxic and induce eosinophilic inflammation in tissues [6,8,25].

### 3.3. Target Organs and Histopathological Findings Based on ANCA Status

In line with the different pathophysiologies of ANCA-positive and ANCA-negative patients, the target organs and histopathological features differ between the two groups.

Patients with negative ANCA results are more likely to develop pulmonary and cardiac involvement. The most common pulmonary involvement is bronchial asthma, which is present in 90% of cases, followed by eosinophilic pneumonia in 38–75% [54,55,56]. Pathological examination of the lungs shows eosinophil infiltration associated with extravascular granulomas and necrotizing vasculitis of capillaries. Eosinophilic pneumonia is common in ANCA-negative patients. Cardiac involvement accounts for 50% of all deaths and is a poor prognostic factor [57,58,59,60]. Cardiac involvement, such as myocarditis, endocarditis, and coronary arteritis, is common in ANCA-negative patients, and the affected lesions often show massive eosinophilic infiltration.

In contrast, ANCA-positive patients frequently develop renal disease and peripheral neuropathy. Renal disease affects approximately 25% of patients and is common among ANCA-positive patients [61,62,63,64]. Pauci-immune crescentic glomerulonephritis is the most typical pathological finding; however, eosinophilic infiltration and granuloma formation are rarely seen [4,6,61]. Membranous nephropathy and membranoproliferative glomerulonephritis can be seen in ANA-negative patients [65]. Peripheral neuropathy due to polyneuritis or mononeuritis occurs in approximately 90% of the patients. Vasculitis in the epineural space is common in the ANCA-positive group, whereas eosinophil infiltration into the lumen of the epineural vessels and endoneurium was prevalent in the ANCA-negative group [66].

Some organ damage, such as skin, gastrointestinal, and central nervous system (CNS) involvement, can be observed regardless of ANCA positivity. Skin lesions are observed in 40–50% of cases. The most common lesion is palpable purpura, followed by livedo, erythema, and blisters [6,67,68,69,70]. Eosinophil-rich leukocytoclastic vasculitis and neutrophilic granulomatosis are often observed [6,70]. Gastrointestinal involvement, such as ulceration and perforation, is observed in approximately 30% of the patients [71,72] and is considered a poor prognostic factor [15,16]. The small intestine is primarily affected, and there is no correlation between the ANCA status and gastrointestinal perforation [71]. Pathological findings range from eosinophil infiltration to vasculitis [4]. CNS involvement, which is often observed at EGPA diagnosis, includes ischemic cerebrovascular lesions, intracerebral hemorrhage and/or subarachnoid hemorrhage, loss of visual acuity, and cranial nerve palsies [73].

Thus, vasculitis leading to inflammation and tissue ischemia is prominent in ANCA-positive patients, whereas eosinophil-related cytotoxicity and vascular occlusion are predominant in ANCA-negative patients [4,8,66,74].

### 3.4. Genetic Differences between ANCA-Positive and ANCA-Negative Patients

The mechanism underlying the pathophysiological differences according to ANCA status has long been unknown. However, a genome-wide association study (GWAS) of EGPA published by the European Vasculitis Genetics Consortium in 2019 revealed that ANCA-positive and ANCA-negative patients have different genetic backgrounds [9] (Figure 3). The study identified several novel genomic loci associated with EGPA. Specifically, the MPO-ANCA-positive group was strongly associated with *HLA-DQ*, whereas the MPO-ANCA-negative group had non-HLA regions such as *GPA33* and *IRF1/IL5* [9]. The *HLA-DQ* locus is shared with MPO-ANCA-positive EGPA and MPO-ANCA-positive ANCA-associated vasculitis (AAV). *GPA33* encodes a cell surface glycoprotein that plays a role in maintaining the barrier function of the intestinal epithelium and bronchial tissue, while *IRF1/IL5* interacts with the regulatory regions of *IL-4*, *IL-5*, and *IRF1* and is associated with increased susceptibility to EGPA, higher eosinophil count, asthma, allergic rhinitis, inflammatory bowel disease, and juvenile idiopathic arthritis [2,6,45,75].

In contrast, genetic variants commonly found in both ANCA-positive and-negative patients included *TSLP*, *BCL2L11*, *CDK6*, *GATA3*, *BACH2*, and *LPP/BCL6* [9]. These variants may contribute to EGPA onset by lowering the activation thresholds of Th2 cells and eosinophils [6]. For example, TSLP is released from stromal and epithelial cells and promotes eosinophilia and Th2 responses. Recently, monoclonal antibodies against TSLP have been shown to reduce asthma exacerbations [76]. *BCL2L11* encodes BIM, essential for regulating cell death and immune homeostasis, *CDK6* encodes a protein kinase involved in cell cycle regulation, *GATA3* encodes a transcription factor expressed by Th2 cells and ILC2s that drive eosinophilic inflammation, *BACH2* encodes a transcription factor that plays an important role in B and T cells, and *LPP/BCL6* is associated with asthma, allergy, and plasma immunoglobulin (Ig) E levels [9,52]. Further genetic studies are needed to identify genetic variants associated with treatment responses and prognosis, as well as new therapeutic targets [8,51,52].

## 4. Development and Performance of the 2022 ACR/EULAR Classification Criteria for EGPA

For the classification of EGPA, the American College of Rheumatology (ACR) classification criteria (1990) [77], Lanham criteria (1984) [78], European Medicines Agency (EMA) algorithm (2007) [79], Chapel Hill Consensus Conference (2012) definitions [1], and European Respiratory Society-endorsed criteria (2017) [80] have long been used [81]; however, the ACR and European Alliance of Associations for Rheumatology (EULAR) jointly developed new classification criteria for EGPA in 2022 (Figure 4) [41]. In addition, the ACR/EULAR committee developed classification criteria for granulomatosis polyangiitis (GPA) and microscopic polyangiitis (MPA) as well [82,83].

To develop these new criteria, EGPA (n = 226) was compared to other types of vasculitis (n = 886), including GPA (n = 300), MPA (n = 291), polyarteritis nodosa (n = 51), non-ANCA-associated small vessel vasculitis that could not be subtyped (n = 51), Behçet’s disease (n = 50), IgA vasculitis (n = 50), cryoglobulinemic vasculitis (n = 34), AAV that could not be subtyped (n = 25), primary central nervous system vasculitis (n = 19), and antiglomerular basement membrane disease (n = 16). EGPA was significantly associated with younger age, lower serum creatinine levels, higher eosinophil counts, and lower positivity for proteinase-3-ANCA than the comparators. MPO-ANCA positivity was not significantly different from that of comparators. Ultimately, MPO-ANCA was not included as an identifier in the new classification criteria [41].

The new criteria showed a sensitivity of 84.9% and a specificity of 99.1% for the classification of EGPA; however, these should not be used as diagnostic criteria [84] and should be applied only after a diagnosis of small- or medium-sized vessel vasculitis has been made. In addition, differential diagnoses that mimic vasculitis must be ruled out before applying these criteria. If the total scores of the seven items are six or more, it is possible to classify a patient as having EGPA [41,84]. However, the histological proof is not always obtained in clinical practice and overlap with HES is likely the case for some patients [5,10,42].

Several studies have validated the performance of the 2022 ACR/EULAR criteria for EGPA, GPA, and MPA. When the 2022 ACR/EULAR criteria were compared with the EMA algorithm, the 2022 criteria successfully reduced the number of “unclassifiable” patients [85]; however, problems with the 2022 criteria were also identified. For example, some patients with EGPA and interstitial lung disease were classified as having MPA in the 2022 criteria, and some patients met both MPA and GPA or MPA and EGPA classification criteria [85]. It is recommended that these classification criteria for EGPA be used in combination with the previous criteria.

## 5. Updated Treatment of EGPA

### 5.1. For Induction of Remission in New-Onset and Relapsing EGPA

Remission induction therapy depends on disease severity. If the patient has organ- or life-threatening manifestations, such as glomerulonephritis, alveolar hemorrhage, cardiac involvement, central nervous system involvement, intestinal ischemia, and peripheral neuropathy, or at least one of the poor prognostic factors defined by the FFS [15,16], a combination of high-dose GCs and cyclophosphamide (CYC) is recommended [12,14], although no studies have demonstrated clear evidence of concomitant use of CYC. A comparison between oral and intravenous administration of CYC showed no clear difference in remission rates in patients with AAV; however, oral administration was associated with an increased risk of leukopenia, infection, and cancer [86], making intravenous administration of CYC preferable. Regarding the number of CYC pulses, in a randomized trial in patients with EGPA and FFS ≥ 1, twelve pulses of CYC had a slightly longer relapse-free survival than six pulses but did not reduce severe relapses [87]. Therefore, the optimal number and duration of CYC pulses have not been established [12,53].

Rituximab (RTX) is considered an alternative to CYC; however, sufficient evidence is lacking [14]. A randomized controlled trial examining the use of RTX in EGPA (REOVAS) has not been published yet; the data reported so far suggest that RTX could be an alternative to CYC, but it did not seem to be superior [88].

In remission induction therapy for new-onset EGPA without organ- or life-threatening manifestations or poor prognostic factors (FFS = 0), treatment with GC alone is recommended [12,14]. As relapses are common once GCs are tapered, other immunosuppressive agents are often combined in these patients. However, evidence supporting the use of immunosuppressive agents is scarce [14]. For the induction of remission in patients with relapsing or refractory EGPA without organ- or life-threatening manifestations or poor prognostic factors (FFS = 0), the use of mepolizumab is recommended [12,14]. The use of mepolizumab is discussed in detail in a following section.

### 5.2. For Maintenance of Remission in EGPA

Cyclophosphamide is not recommended for the maintenance of remission due to side effects, such as bladder cancer and infertility. Instead, in patients with organ- or life-threatening manifestations or poor prognostic factors, methotrexate, azathioprine, mepolizumab, or RTX should be considered in combination with GCs after remission induction with CYC [12,14]. In patients without organ- or life-threatening manifestations, or poor prognostic factors after remission induction, mepolizumab is recommended at the time of relapse [12,14]. The results of the REOVAS trial are awaited, particularly in ANCA-positive patients.

### 5.3. Mepolizumab for EGPA

Mepolizumab is a humanized monoclonal antibody that binds to and neutralizes IL-5 [45]. It has been approved for severe asthma and EGPA, and its approval for EGPA is based on the MIRRA study [18].

The MIRRA study was conducted in 2017 to evaluate the efficacy and safety of mepolizumab (subcutaneous injection of 300 mg/4 weeks) in combination with GC therapy in patients with relapsed or refractory EGPA. Of note, enrolled patients had relapsed while receiving at least 7.5 mg of GCs or had failed to achieve remission after 3 months or more of standard induction of remission before study entry; however, patients with organ or life-threatening manifestations at the time of study entry were excluded. Moreover, the mean disease duration was approximately 5 years, ANCA positivity was as low as 10%, the mean eosinophil count was approximately 170 cells/μL, and the prevalence of peripheral neuropathy was approximately 40%, indicating that these patients did not have the active severe disease.

The study results demonstrated that, compared with a placebo, mepolizumab treatment significantly reduced eosinophil counts, induced a higher proportion of patients in remission, and maintained remission for a longer duration with reduced relapses. In addition, the GC dose was successfully reduced in the mepolizumab group, and 18% of patients in the mepolizumab group achieved GC-free remission during weeks 48 and 52. Regarding safety, local injection site reactions were similar, but headaches were more common in the mepolizumab group than in the placebo group [18].

Based on these results, mepolizumab is currently recommended for the induction of remission in patients with relapsing or refractory EGPA without organ- or life-threatening manifestations and for the maintenance of remission after induction of remission [12,14]. Following the MIRRA study, multiple observational studies and case series on mepolizumab have been reported.

It was predicted that mepolizumab would have greater efficacy in ANCA-negative EGPA, according to the above-mentioned GWAS showing that genetic variants in *IL-5* were associated only with ANCA-negative EGPA [9]; however, the efficacy of mepolizumab was not associated with ANCA positivity in the MIRRA study [18]. This debate is still ongoing; a report from a European multicenter observational study showed that mepolizumab was more effective in ANCA-negative patients than in ANCA-positive patients [89], whereas other studies reported that mepolizumab was effective regardless of ANCA positivity [90,91]. Therefore, it may be too early to draw a sound conclusion only from these studies.

Although evidence for the use of mepolizumab against acute severe EGPA is lacking due to the exclusion criteria of the MIRRA study, a retrospective observational study compared the efficacy and safety of mepolizumab with an intravenous CYC pulse (IVCY) as a remission induction therapy for severe EGPA [92]. The efficacy of mepolizumab in controlling the disease activity was comparable to that of IVCY, and the GC-sparing effect was better in the mepolizumab group. Mepolizumab had no new safety signals [92]. The authors of this study reported a case of EGPA with acute organ-threatening manifestations, such as gastrointestinal lesions and peripheral neuropathy, in which mepolizumab was introduced in combination with GC from the induction phase of remission, and GC was ultimately withdrawn [93]. Thus, weaning of concomitant GC therapy may be an achievable therapeutic goal of mepolizumab.

Several attempts have been made to treat organ- or life-threatening manifestations of EGPA with mepolizumab, such as cardiomyopathy and gastrointestinal involvement [19,94], the efficacy of which is yet to be determined. However, a growing number of case reports and case series have shown that symptoms related to peripheral neuropathy were ameliorated and nerve conduction velocity tests improved with mepolizumab [95,96,97]. This appears promising, but the supporting evidence is currently insufficient.

### 5.4. Other IL-5-Targeting Agents and Biologics for EGPA

When these therapies are not sufficiently effective, other IL-5-targeting agents, plasma exchange, or intravenous immunoglobulin (IVIG) may be considered [12]. Other IL-5-targeting antibodies besides mepolizumab include reslizumab and benralizumab, both of which are indicated for severe asthma [45]. Reslizumab is an anti-IL-5 monoclonal antibody that blocks eosinophil survival and proliferation. Benralizumab is an IL-5 receptor alpha-directed cytolytic monoclonal antibody that inhibits eosinophil differentiation and maturation in the bone marrow. It can also bind to the FCγRIIIa on NK cells, macrophages, and neutrophils via its afucosylated Fc domain, potently inducing antibody-dependent cell-mediated cytotoxicity (ADCC). Benralizumab almost completely depletes eosinophils through these mechanisms [21].

The efficacy of reslizumab and benralizumab in EGPA has not been tested in a randomized controlled study but has been demonstrated in several case reports. Reslizumab may have a GC-sparing effect on EGPA [98,99]. In a case series, benralizumab was effective in patients with mepolizumab-refractory EGPA with cardiac and neuropathic complications [100]. Large-scale trials are currently underway to evaluate the efficacy and safety of benralizumab for EGPA [19,53,101].

Dupilumab, an anti-IL-4/IL-13 receptor antibody, may be used for severe asthma and ENT manifestations associated with EGPA [102,103]. However, dupilumab administration sometimes leads to an increase in eosinophils and worsens EGPA [102,104,105,106].

### 5.5. Intravenous Immunoglobulin (IVIG) for EGPA

Several observational studies have shown the efficacy of IVIG for peripheral neuropathy on EGPA [67,107,108,109]. In addition, an increase in the left ventricular ejection fraction and recovery of myocardial function has been observed in some patients with cardiac involvement [67,108,109]. A meta-analysis evaluating the efficacy of IVIG in patients with active AAV, including EGPA, showed significant reductions in BVAS, ANCA titers, and C-reactive protein levels within 6 months of IVIG administration [110]. Further studies are required to verify the efficacy of IVIG for EGPA.

## 6. Unmet Clinical Needs on EGPA

Several challenges remain for the diagnosis, treatment, and management of EGPA. The establishment of diagnostic criteria for EGPA and identification of disease-specific biomarkers that can monitor disease activity and treatment efficacy are desirable [12]. In particular, diagnosing EGPA while using mepolizumab or benralizumab for severe asthma may be difficult. However, whether these IL-5-targeting agents for severe asthma suppress EGPA development remains unclear. Whether the treatment of EGPA should be tailored according to ANCA status remains inconclusive.

In addition, the optimal initial GC dose and methods for tapering the GC dose have not yet been fully established. In remission induction therapy for MPA and GPA, a reduced-dose GC regimen is almost as effective as a standard-dose regimen and has fewer side effects [111,112,113]. In addition, avacopan, a C5a receptor inhibitor, has emerged as an alternative to high-dose GC therapy for MPA and GPA but not for EGPA [114]. Thus, there is insufficient evidence regarding the optimal use of GC during both the remission induction and maintenance phases of EGPA.

Moreover, the MIRRA study did not include acute severe patients with life- or organ-threatening manifestations [18]; therefore, there are uncertainties regarding the use of mepolizumab for acute severe diseases [19]. Other unresolved questions include (1) whether 300 mg/4 weeks instead of 100 mg/4 weeks is necessary for all patients with EGPA [94]; (2) whether mepolizumab can be combined with immunosuppressive agents, such as RTX [115]; (3) whether mepolizumab can be combined with other biological agents, such as anti-IgE antibody and anti-IL-4/IL-13 receptor antibody [116]; (4) whether mepolizumab suppresses organ damage over the long term; and (5) whether mepolizumab ultimately improves the prognosis for EGPA. Because half of the mepolizumab group in the MIRRA study experienced relapse, it is essential to establish biomarkers to predict the efficacy of mepolizumab.

## 7. Conclusions

Although EGPA is a relatively rare disease, its diagnosis and treatment are constantly progressing owing to advances in basic and clinical research and the introduction of new biological agents. Evidence-based treatment recommendations for remission induction and remission maintenance were proposed, providing us with a rationale for treatment. However, several challenges remain unaddressed, such as the optimal initial GC dose and use of mepolizumab for acute severe EGPA. Therefore, it is essential to promote a multidisciplinary approach to pursue the best treatment for each patient.

## Figures and Tables

**Figure 1 jcm-12-05996-f001:**
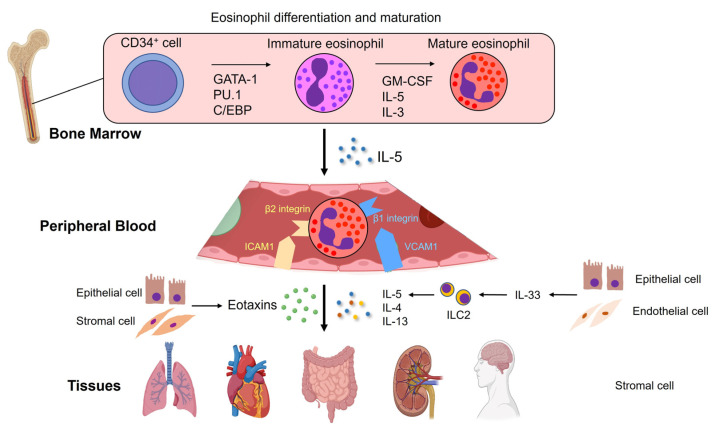
**Eosinophils matured in the bone marrow migrate to the peripheral blood and are mobilized to tissues.** Eosinophils are differentiated from CD34-positive hematopoietic stem cells in the bone marrow by transcription factors, such as GATA-1, PU.1, and C/EBP, and mature in the presence of granulocyte macrophage-colony-stimulating factor (GM-CSF), interleukin (IL)-5, and IL-3. Mature eosinophils are released from the bone marrow into the peripheral blood by IL-5. In the peripheral blood, eosinophils circulate, roll around on the endothelial cells, and leak out from blood vessels by binding eosinophil surface adhesion molecules (β1 and β2 integrins) to vascular cell adhesion molecule-1 (VCAM1) and intercellular adhesion molecule-1 (ICAM1) expressed on vascular endothelial cells. Type 2 innate lymphoid cells (ILC2s) abundantly produce IL-5 in response to IL-33 from epithelial and endothelial cells during allergic inflammation. Eotaxin produced by epithelial and stromal cells cooperates with IL-5 to selectively mobilize eosinophils to sites of eotaxin expression. In addition, cytokines such as IL-4, IL-5, and IL-13 and eotaxins also contribute to the migration of eosinophils into inflammatory tissues.

**Figure 2 jcm-12-05996-f002:**
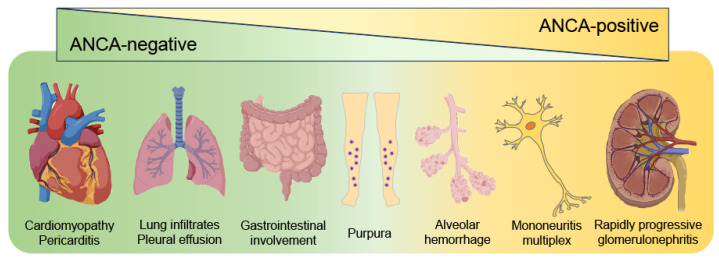
**Clinical manifestations differ between ANCA-positive and ANCA-negative EGPA.** In antineutrophil cytoplasmic antibody (ANCA)-positive cases of eosinophilic granulomatosis with polyangiitis (EGPA), rapidly progressive glomerulonephritis and peripheral neuropathy are more likely to occur. In contrast, ANCA-negative cases are more likely to develop cardiac and pulmonary involvement. Gastrointestinal lesions, purpura, and alveolar hemorrhage are observed regardless of ANCA status.

**Figure 3 jcm-12-05996-f003:**
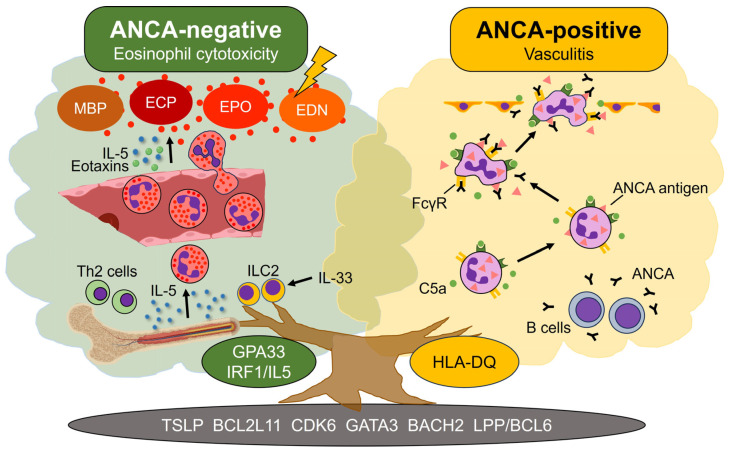
**Genetic differences between ANCA-positive and ANCA-negative EGPA lead to different pathophysiology.** Antineutrophil cytoplasmic antibody (ANCA)-negative eosinophilic granulomatosis with polyangiitis (EGPA) is genetically associated with *GPA33* and *IRF1/IL5*. In this group, the interleukin (IL)-5-eosinophil axis plays a central role in the pathogenesis. Specifically, IL-5 produced by T helper 2 (Th2) cells and type 2 innate lymphoid cells (ILC2s) promotes eosinophil maturation and proliferation in the bone marrow and extravasation from the peripheral blood to the site of inflammation in concert with eotaxins. Eosinophils that reach to the tissues initiate degranulation, and cytoplasmic granules, such as major basic protein (MBP), eosinophil cationic protein (ECP), eosinophil peroxidase (EPO), and eosinophil-derived neurotoxin (EDN), exhibit cytotoxicity, causing eosinophilic inflammation. In contrast, ANCA-positive EGPA is genetically associated with *HLA-DQ*. In this group, circulating neutrophils are primed by inflammatory cytokines and complement C5a. Primed neutrophils expose cytoplasmic ANCA antigens to the cell surface, allowing circulating ANCA produced by activated B cells to bind to ANCA antigens. The Fc fragments of ANCA further activate neutrophils via Fcγ receptor (FcγR), which result in the overproduction of reactive oxygen species and the formation of neutrophil extracellular traps. Thus, neutrophil-mediated vasculitis represents the main pathology in ANCA-positive EGPA. Both ANCA-positive and ANCA-negative EGPA share genetic variants in *TSLP*, *BCL2L11*, *CDK6*, *GATA3*, *BACH2*, and *LPP/BCL6*, which may lower the activation threshold of Th2 cells and eosinophils.

**Figure 4 jcm-12-05996-f004:**
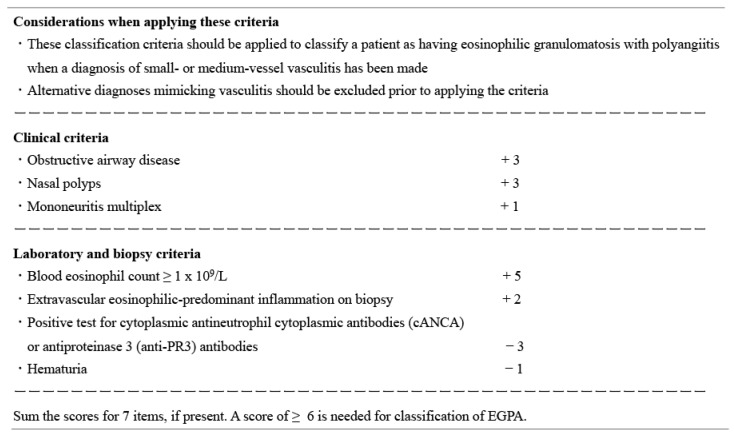
The 2022 American College of Rheumatology/European Alliance of Associations for Rheumatology classification criteria for eosinophilic granulomatosis with polyangiitis (EGPA).

## Data Availability

Not applicable.

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
