# Peer review of "Eosinophilic Granulomatosis with Polyangiitis: Latest Findings and Updated Treatment Recommendations"

_jcm, 2023, doi:10.3390/jcm12185996_

Round 1

Reviewer 1 Report

In this manuscript, Watanabe and Hashimoto present a throughout review of eosinophilic granulomatosis with polyangiitis (EGPA). The manuscript is well-written and very informative. I have only the following suggestions:

1. Figure 1 would benefit from a few additions such as labeling the CD34+ cells, the immature and mature eosinophils, and adding that endothelial and epithelial cells when activated by infections produce IL-33.

2. In lines 142 to 144 the authors mention that ILC2 could be a potential therapeutic target for EGPA because they produce IL-5. What about the IL-5 that is produced by Th2 cells?

3. Regarding EETosis, the authors should discuss the limitations of the extracellular traps that are induced by PMA, as the effect of PMA does not seem to replicate any physiological situation. Are there any studies of EETOsis in EGPA using a live cell imaging microscope?

4. It would be helpful to list or include on a table the 2022 ACR/EULAR classification criteria for EGPA. 

5. They mention that is essential to establish biomarkers to predict the efficacy of mepolizumab, but biomarkers are also needed for diagnosis, and for disease activity, and to predict efficacy and measure the response to many other medications, including steroids, and cyclophosphamide. 

Author Response

We appreciate your insightful comments. We revised our manuscript based on the reviewer’ comments. We highlighted the changes in red. Point-by-point responses to the reviewer’ comments are as follows.

1. Figure 1 would benefit from a few additions such as labeling the CD34+ cells, the immature and mature eosinophils, and adding that endothelial and epithelial cells when activated by infections produce IL-33.

Response: Thank you very much for pointing it out. Based on your comments, we revised Figure 1 and figure legends (Line 112).

2. In lines 142 to 144 the authors mention that ILC2 could be a potential therapeutic target for EGPA because they produce IL-5. What about the IL-5 that is produced by Th2 cells?

Response: Thank you very much for your comments. Based on your comments, we revised the manuscript (Line 142).

3. Regarding EETosis, the authors should discuss the limitations of the extracellular traps that are induced by PMA, as the effect of PMA does not seem to replicate any physiological situation. Are there any studies of EETOsis in EGPA using a live cell imaging microscope?

Response: Thank you very much for your excellent comments. To our knowledge, any studies have shown EETosis using a live cell imaging. Based on your comments, we revised the manuscript (Line 182-184).

4. It would be helpful to list or include on a table the 2022 ACR/EULAR classification criteria for EGPA. 

Response: We appreciate your comments. We added the 2022 classification criteria in the manuscript (Figure 4).

5. They mention that is essential to establish biomarkers to predict the efficacy of mepolizumab, but biomarkers are also needed for diagnosis, and for disease activity, and to predict efficacy and measure the response to many other medications, including steroids, and cyclophosphamide. 

Response: We agree with your comments. We revised the manuscript (Line 471).

Reviewer 2 Report

The authors have provided a detailed and up-to-date synopsis of the current state of play surrounding the diagnosis, pathophysiology and treatment of EGPA which, as they correctly state, has evolved rapidly in recent years. 

I have read the well-structured manuscript with interest and have learned from it. The illustrations are appropriate and excellent. The explanations and messages appear accurate and well-referenced. I have a few suggestions to make but no major criticisms of this work which I feel makes a timely and valuable contribution to the management of this condition.

1 Would the authors agree to introduce the dose regime and administration route for mepolizumab in section 5.3 rather than leaving it until section 6?

2 I recommend an expanded 'conclusion' section, perhaps incorporating the key messages that the authors wish to share with their readers.

Author Response

We appreciate your insightful comments. We revised our manuscript based on the reviewer’ comments. We highlighted the changes in red. Point-by-point responses to the reviewer’ comments are as follows.

1 Would the authors agree to introduce the dose regime and administration route for mepolizumab in section 5.3 rather than leaving it until section 6?

Response: Thank you very much for pointing it out. Based on your comments, we added the dose and administration route of mepolizumab in section 5.3 (Line 395).

2 I recommend an expanded 'conclusion' section, perhaps incorporating the key messages that the authors wish to share with their readers.

Response: Thank you very much for your suggestion. Based on your comments, we expanded the conclusion section (Line 497-500).